# Morphometric Relationships between Length and Weight of 109 Fish Species in the Caribbean Sea (French West Indies)

**DOI:** 10.3390/ani13243852

**Published:** 2023-12-14

**Authors:** Kélig Mahé, Jérome Baudrier, Angela Larivain, Solène Telliez, Romain Elleboode, Elise Bultel, Lionel Pawlowski

**Affiliations:** 1IFREMER, Unité HMMN, 150 quai Gambetta, BP 699, 62321 Boulogne-sur-Mer, France; solene.telliez@ifremer.fr (S.T.); romain.elleboode@ifremer.fr (R.E.); 2IFREMER, Délégation pour les Antilles, Station de Martinique, Unité Biodivenv, 97231 Le Robert, France; jerome.baudrier@ifremer.fr; 3IFREMER, UMR DECOD, LTBH, 56100 Lorient, France; angela.larivain@ifremer.fr (A.L.); lionel.pawlowski@ifremer.fr (L.P.); 4IFREMER, Unité HISSEO, 29280 Brest, France; elise.bultel@ifremer.fr

**Keywords:** GLM, L–W relationship, sexual dimorphism, fork length, temporal effect, Guadeloupe, Martinique

## Abstract

**Simple Summary:**

Poor quality of biological information, such as the relationship between length and weight parameters, could be a source of variability with a significant impact on stock assessment results. This relationship between the length and weight of individuals could differ between males and females as well as two separated islands and/or linked to the reproduction period according the studied species.

**Abstract:**

In total, 109 fish species (24,996 individuals) were sampled around Guadeloupe and Martinique Islands from October 2021 to September 2022 to estimate the morphometric relationships between total length and weight (Length Weight Relationship: LWR) of each fish species according to potential spatial, temporal and sex differences. Of these species, this is the first time that the LWR was estimated in the Atlantic Ocean for 16 species. There is a significant relationship between length and weight for all tested species. For 83 tested species, the sex effect on the LWR showed significant sexual dimorphism for 24 species. Additionally, a link between the temporal effect and the reproduction period was tested for 68 species, of which 35 presented significant differences relative to the annual quarter of sampling. Finally, the geographical effect (i.e., the difference between samples from around Guadeloupe Island and those from Martinique Island) was significant for 60 species. This island effect was significant for 25 species.

## 1. Introduction

Biological information, such as body length and weight, is a pre-requisite in fishery sciences to build stock assessments based on length-at-age and weight-at-age data [1]. However, both length and weight data are not necessarily acquired during sampling. Consequently, the characterization of the length–weight relationship (LWR) allows readily usable estimates of each of these values. Researchers using morphometric data may not have enough information should an accurate conversion between different metrics be unavailable [2,3]. Moreover, this relationship is a sustainable proxy for “fatness” and “general well-being”, like the condition factor [4,5,6]. The exponential function is the main method to characterize the LWR of fish, which depends on environmental conditions and the physiological state of the individual [1,4,7,8]. The fatness and consequently the outline shape of the fish species can be analyzed from the growth coefficient, which presents values ranging from 2.5 to 4 [1,9,10,11,12]. To weigh a fish, total weight (W_T_) is generally the most commonly used metric. Occasionally, the gutted weight can also be used; however, there is often a conversion factor to obtain the proportion of total weight that is represented by this proxy. Conversely, when measuring the length of a fish, there are many different potential lengths according to the anatomy of sampled species, especially the shape of the tail (i.e., with or without tail filaments). However, the two main metrics of measuring fish for research, resource management and regulation purposes are total length (TL: measured from the tip of the jaw or the snout to the farthest tip of the tail with the tail squeezed) and fork length (FL: measured from the tip of the jaw or the snout to the center of the fork in the tail).

The West Indies extend northward from South America in a long volcanic arc of over a dozen major islands and hundreds of small islets (from the Dominican Republic in the north to the Grenadines in the south of the Lesser Antilles). These islands are among the 34 identified global biodiversity hotspots [13,14] and are home to a high number of endemic genera and species [15]. The islands are of various sizes, and the largest is Guadeloupe Island at 1706 km^2^. The length–weight relationships of fish species belonging to this biodiversity hotspot are relatively unknown. The literature provides sparse LWR for some local species from studies in adjacent regions, but generally from different marine environments several hundreds if not of thousands of kilometers distant (e.g., Florida or Brazil). These great distances likely make the use of published biological parameters unreliable to tackle the growth dynamics of local species. In the West Indies, a few specific studies have focused on a single species, such as the lionfish [16,17,18,19,20,21,22], which is an invasive species. Some papers have reported the values for several species, but these are often several years or even decades old (i.e., from the 1970s and 1980s). Consequently, this study focused on the length–weight relationships for 109 fish species caught in the waters of the French West Indies. There are four French islands (Guadeloupe, Martinique, Saint Barth and Saint Martin), and the two main islands are in the middle of the long arc with Guadeloupe Island in the north and Martinique Island in the south, separated by around 190 km, with Dominica Island in between (Figure 1). The potential influence of factors, such as sampling period (i.e., sampling quarter), sampling location/island (i.e., Guadeloupe and Martinique Islands) and physiological effect (i.e., the potential sexual dimorphism between males and females), on length–weight relationships was evaluated.

## 2. Materials and Methods

Among all species, only those with a minimum sample number of 10 individuals were analyzed. This amount often provides a sufficiently precise relationship, but this can be completed with further sampling in the future. The results are presented in two groups. One group includes the species showing higher precision in the analysis (i.e., specimen number > 30). For the other group, only preliminary results are presented in Appendix A due to the small number of specimens (i.e., number < 30). Consequently, 109 fish species were analyzed (*Acanthurus bahianus*, *A. chirurgus*, *A. coeruleus*, *A. tractus*, *Albula vulpes*, *Balistes capriscus*, *B. vetula*, *Canthidermis sufflamen*, *Melichthys niger*, *Xanthichthys ringens*, *Tylosurus crocodilus*, *Caranx bartholomaei*, *C. crysos*, *C. latus*, *C. lugubris*, *C. ruber*, *Chloroscombrus chrysurus*, *Decapterus punctatus*, *Selar crumenophthalmus*, *Selene brownii*, *S. vomer*, *Seriola dumerili*, *S. rivoliana*, *Trachinotus goodei*, *Chaetodipterus faber*, *Chaetodon capistratus*, *C. sedentarius*, *C. striatus*, *Dactylopterus volitans*, *Diodon holocanthus*, *Elops saurus*, *Diapterus auratus*, *Eucinostomus argenteus*, *Gerres cinereus*, *Anisotremus surinamensis*, *Brachygenys chrysargyreum*, *Haemulon aurolineatum*, *H. bonariense*, *H. carbonarium*, *H. flavolineatum*, *H. parra*, *H. plumierii*, *H. sciurus*, *H. striatum*, *Holocentrus adscensionis*, *H. rufus*, *Myripristis jacobus*, *Neoniphon marianus*, *Kyphosus sectatrix*, *Bodianus rufus*, *Clepticus parrae*, *Halichoeres radiatus*, *Lachnolaimus maximus*, *Etelis oculatus*, *Lutjanus analis*, *L. apodus*, *L. buccanella*, *L. cyanopterus*, *L. griseus*, *L. jocu*, *L. mahogoni*, *L. synagris*, *L. vivanus*, *Ocyurus chrysurus*, *Pristipomoides macrophthalmus*, *Rhomboplites aurorubens*, *Malacanthus plumieri*, *Aluterus monoceros*, *A. scriptus*, *Cantherhines macrocerus*, *C. pullus*, *Mulloidichthys martinicus*, *Pseudupeneus maculatus*, *Gymnothorax moringa*, *Acanthostracion polygonius*, *Lactophrys triqueter*, *Polydactylus oligodon*, *Polydactylus virginicus*, *Holacanthus tricolor*, *Abudefduf saxatilis*, *Heteropriacanthus cruentatus*, *Priacanthus arenatus*, *Scarus iseri*, *S. taeniopterus*, *S. vetula*, *Sparisoma aurofrenatum*, *S. chrysopterum*, *S. frondosum*, *S. rubripinne*, *S. viride*, *Larimus breviceps*, *Umbrina coroides*, *Scomberomorus regalis*, *Pterois volitans*, *Scorpaena plumieri*, *Alphestes afer*, *Cephalopholis cruentata*, *Cephalopholis fulva*, *Epinephelus guttatus*, *E. striatus*, *Paranthias furcifer*, *Rypticus saponaceus*, *Archosargus rhomboidalis*, *Calamus bajonado*, *C. calamus*, *C. penna*, *C. pennatula*, *Sphyraena barracuda* and *Synodus intermedius*) (Appendix A). The total sample number per species varied from 10 individuals for *S. regalis* to 1501 individuals for *O. chrysurus*. A total of 24,996 marine fishes caught around Guadeloupe and Martinique Islands were sampled from October 2021 to September 2022 (except the first quarter of 2022). These 2 French islands are located in the archipelago of West Indies Islands, separating the Atlantic Ocean from the Eastern part of the Caribbean Sea. Guadeloupe and Martinique Islands are located 187 km apart with the Dominica Island between these two French islands (Figure 1). Consequently, the sampling from two islands could be interpreted as a potential geographical factor that could influence the morphological characteristics of the fish. All specimens were obtained from landings of fishing boats with commercial and non-commercial species, covering all lengths, including those below the legal size with authorization from the French government to perform the first large sampling for stock assessment. In the non-commercial fishes, there were discarded species due to low price and contaminated by ciguatera poisoning. All specimens were weighed (one individual weight: total, W_T_ ± 0.1 g) and measured (two individual lengths: total, TL ± 0.5 cm and fork, FL ± 0.5 cm). Sex was determined by macroscopic gonad observation for 83 species, but these biological data were not observed for 26 species. All specimens were analyzed fresh. For 26 species, the gonad observation was too difficult to obtain results with a high accuracy. Among the 24,996 individuals, 32 families belonging to Actinopterygii were represented (Acanthuridae, Albulidae, Balistidae, Belonidae, Carangidae, Chaetodontidae, Dactylopteridae, Diodontidae, Elopidae, Gerreidae, Haemulidae, Holocentridae, Kyphosidae, Labridae, Lutjanidae, Malacanthidae, Monacanthidae, Mullidae, Muraenidae, Ostraciidae, Polynemidae, Pomacanthidae, Pomacentridae, Priacanthidae, Scaridae, Sciaenidae, Scombridae, Scorpaenidae, Serranidae, Sparidae, Sphyraenidae and Synodontidae). Among these Actinopterygii families, several families had a large number of species, such as Carangidae (13 species, n = 2086), Lutjanidae (13 species, n = 4927), Haemulidae (10 species, n = 3344), Scaridae (8 species, n = 3885) and Serranidae (7 species, n = 1117) (Appendix A).

All data were plotted to identify and delete the potential outliers. After this preliminary control step, the data were used to characterize the LWR for each species (Equation (1)):W_T_ = a·TL^b^
(1)

The body shape coefficient is expressed as a, and the growth coefficient is represented by b [1,4,23]. The difference between this observed growth coefficient for each fish species and the theoretical value (b = 3.0) was tested with a one-sample *t*-test at a significance level of 0.05 to verify whether it was significantly different from the theoretical value of 3.0. The relationship between the two length measurements (Length–Length Relationship [LLR]) is also analyzed (Equation (2)):TL = c·FL + d(2)

A complete generalized linear model (GLM) was used to tested the potential effects of the explanatory variables sex (S; females and males; 83 tested species), sampling period (Q; Quarter 4 2021; Quarter 2 2022 and Quarter 3 2022; 68 tested species) and geographical location/sampling island (G; 13506 individuals caught around Guadeloupe Island and 11490 individuals caught around Martinique Island) (Appendix A) on the LWR of each species (Equation (3)):log W_T_ ~ (log TL) + (log TL)·S + (log TL)·Q + (log TL)·G + ε(3)

For hermaphrodite species (i.e., change sex from female to male; Appendix A), the difference between males and females for these species was directly influenced by the difference of length and age ranges between sexes. Consequently, for these hermaphrodite species, represented by 4 families Labridae, Pomacanthidae, Scaridae and Serranidae (*Bodianus rufus*, *Clepticus parrae*, *Lachnolaimus maximus*, *Holacanthus tricolor*, *Scarus iseri*, *Scarus taeniopterus*, *Scarus vetula*, *Sparisoma aurofrenatum*, *S. rubripinne*, *S. viride*, *Alphestes afer*, *Cephalopholis cruentata*, *C. fulva*, *Epinephelus guttatus* and *E. striatus*), the potential effect of sex was not tested.

The Fulton’s condition factor (Kc) and the relative condition factor (Kn) [4] were applied to compare the observed total weight at the given length to the predicted total weight from the LWR (Equations (4) and (5)):Kc = 100·W_T_/TL^3^(4)
Kn = W_T_/(a·TL^b^)(5)

All statistical analyses (with significant effect at *p* < 0.05) were carried out in the R statistical environment (version 4.0.2), specifically using the CAR and GGPLOT2 packages [24,25].

## 3. Results

The number of measured specimens and all information (mean ± standard deviation; the minimum and the maximum) for the three individual morphometric parameters (total length, fork length and total weight) are shown in Appendix A for each fish species. Measured total length ranged from 6.5 cm (*B. capriscus*) to 119 cm (*S. barracuda*). The individual total weight range ranged from 5 g (*H. aurolineatum*) to 8750 g (*C. pennatula*) (Appendix A). However, all LWR values are presented because there were either no previous results or very old data for several species. Among the 109 species, this study is the first time that the LWR was estimated for 16 species (*A. bahianus*, *H. striatum*, *N. marianus*, *K. sectatrix*, *B. rufus*, *C. parrae*, *H. radiates*, *C. pullus*, *P. oligodon*, *S. taeniopterus*, *S. vetula*, *S. aurofrenatum*, *S. frondosum*, *P. furcifer*, *R. saponaceus* and *S. intermedius*) and that the LLR was calculated for 7 species (*A. bahianus*, *H. striatum*, *G. moringa*, *A. polygonius*, *S. taeniopterus*, *S. aurofrenatum*, *S. frondosum*) in the Atlantic Ocean for more than 6 individual specimens (Table 1 and Appendix A).

The LWR for each species consistently showed a significant correlation between two morphological parameters (Table 1 and Appendix A). The relationship between the coefficients of LWR also showed a significant correlation between body shape coefficient and the growth coefficient (Figure 2). Only nine sampled species exhibited a b coefficient value equal to 3.0. Conversely, 65 species showed that the body became more elongated throughout life (i.e., b < 3.0), while individuals became thicker (i.e., b > 3.0) over the lifespan in 36 species.

First, the influence of sex was estimated for the 83 species. The results (i.e., the slopes of the LWR) identified significant differences between females and males for 24 species (i.e., 29% of all species: *M. niger*, *C. crysos*, *S. crumenophthalmus*, *S. rivoliana*, *C.striatus*, *D. volitans*, *G. cinereus*, *H. aurolineatum*, *B. chrysargyreum*, *H. adscensionis*, *H. rufus*, *K. sectatrix*, *L. buccanella*, *L. vivanus*, *C. macrocerus*, *P. maculatus*, *A. polygonius*, *H. tricolor*, *S. taeniopterus*, *S. aurofrenatum*, *S. chrysopterum*, *S. rubripinne*, *S. viride* and *E. guttatus*; Table 1 and Appendix A). For each species showing a significant difference between the sexes, the LWR was calculated for each sex (Appendix A). Second, a temporal effect (i.e., the sampling quarter on the slopes of the LWR) was detected for 35 species (i.e., 51% of 68 tested species). Finally, geographical differences in the LWR between Guadeloupe and Martinique Islands were tested for 60 species, and this effect was significant for 25 species (42%). For 103 species where the fork length was measured, the total length–fork length relationship was always significant (*p* < 0.05; Table 1). To understand the difference between species and/or potential explanatory variables (sex, sampling quarter and island), condition factors (Kc and Kn) were calculated (Figure 3 and Figure 4). First, comparing Kn and Kc conditions factors on the same dataset, the trend for each factor was not comparable to another factor (Figure 3). Second, to analyze the condition factor according to geographical location/sampling island, sex, and sampling period, only Kn was used (Figure 4) The condition factor Kn was greater for females than males for 61 species, while 22 species presented the opposite trend (Figure 3). There is no clear trend among sampling quarters. Finally, the condition factor of fish from Martinique Island was often higher than that for those from Guadeloupe Island (i.e., 49 species vs. 22 species; Figure 4).

## 4. Discussion

Morphometric relationships between length and weight were established for 109 fish species (n = 24,996 individuals). This variety of species is substantially higher than those observed in the temperate waters (i.e., Northeastern Atlantic Ocean [26], 45 species for 31,167 individuals) but closer to those observed in other warm water areas (i.e., the Gulf of Mexico [17], 174 fish species for 7503 individuals; Réunion Island (Indian Ocean) [27], 123 fish species for 10,218 individuals). This number is within the ranges of the count of fish species observed in the fish assemblage of the Caribbean arc, where between 63 and 167 fish species were reported per island [28,29]. These studies identified all commercial and non-commercial species without a minimum threshold number, whereas this value was fixed at 10 for this LWR study. However, this study on length–weight relationships showed that fish species diversity is higher than observed in previous studies on biological information in the Caribbean Sea (n = 37 species, Gulf of Salamanca, Columbia [30]; n = 20 species, Gulf of Salamanca [18]; n = 36 species, deep-sea fishes in Colombia [20]; n = 66 species, delta of the Atrato River, Colombia [21]; n = 22 species, freshwater fish in Venezuela [22]). Our results showed that the body length–weight relationship for all tested species was significant, which is consistent with previous studies carried out in the Northeastern Atlantic Ocean [26,31,32], Greek waters [33], the Persian Gulf [34], the Aegean Sea [35] and the Indian Ocean [27]. These results confirmed that, even with small numbers of sampled individuals, there is a significant relationship between length and weight, which allows us to switch from one morphological metric to the other and vice versa. For many species, this study either presented the first length–weight relationship for the Caribbean Sea (i.e., LWR only available for other geographical areas), or updated the LWR in the Caribbean Sea, which may be several decades old. Finally, for 16 species (*A. bahianus*, *H. striatum*, *N. marianus*, *K. sectatrix*, *B. rufus*, *C. parrae*, *H. radiates*, *C. pullus*, *P. oligodon*, *S. taeniopterus*, *S. vetula*, *S. aurofrenatum*, *S. frondosum*, *P. furcifer*, *R. saponaceus* and *S. intermedius*), this was the first time that a length–weight relationship has been identified in the scientific literature, providing basic data for monitoring fish populations. Among these species, 8 species had a higher precision sampling number (n > 30; Appendix A). Similarly, the LLR showed significant correlations between fork and total lengths, which corroborated previous studies [36,37]. For all species with different tail shape, this result allows us to switch from one length to the other and vice versa. As with the LWR analysis, this is the first time that the LLR is calculated for 7 species (*A. bahianus*, *H. striatum*, *G. moringa*, *A. polygonius*, *S. taeniopterus*, *S. aurofrenatum* and *S. frondosum*) for which total length can be transformed into fork length or vice versa without introducing bias into the measurements. Among 109 analyzed species, 72 species had more than 30 individuals measured and a good coverage of the known size range for each species. For the remaining 37 species, results are preliminary and can be updated in the future. In this study, a value of 3.0 (i.e., isometric growth) for the allometric coefficient b was observed for only 9 species (8.2% of the total number of species, Figure 2). The b values ranged between 2.0 and 4.0, confirming the previous results for the large number of species [38]. For this reason and the correlation between two coefficients of LWR, two condition factors were used to identify the sources of difference among LWR. Traditionally, the Fulton’s condition factor (Kc) based on isometric growth (b = 3.0) is applied, but the relative condition factor (Kn), defined by allometric growth (b < 3 or b > 3), is more representative for the wild fish species. Comparing the trend for both condition factors, there were several differences among these 109 species. As Kn directly uses the values of LWR coefficients extracted from the sampling data, it will be preferable to use the allometric condition factor Kn instead of the Fulton’ isometric condition factor (Kc) [4,39]. The value of Kn is compared to a threshold value of 1.00, above which fish are in good health and grow in favorable conditions, limiting the stress of individuals (i.e., good range of environmental conditions, available prey and low density of predators) [4,40,41,42,43]. Within the sampled species, there are several potential factors that main explain differences in the LWR, such as the physiological aspect (ontogenic effect, sexual dimorphism, reproduction period vs. other months) and/or the environment (difference of habitats between islands) [26].

The LWR was first tested for a physiological effect on the difference between males and females. Among the 109 species analyzed, the influence of sex was estimated for 83 species, and 24 of those species presented significant sexual dimorphism (29%). According to the value of relative condition factor (Kn), the sexual dimorphism showed that females are often heavier than males of the same length as previously shown for Northeastern Atlantic Ocean species [26,44]. In the present study, the effect of sampling quarter was investigated because it could be directly related to reproduction period (gonad development and spawning period) [6,45,46]. The sampling period showed no clear trend (significant effect for 35 species among 68 tested species). For the species with a significant effect, the sampling quarter could be linked to the reproduction period or sexual rest period (i.e., for each species, Appendix A). The females were heaviest just before and during the spawning period, particularly due to the degree of gonad development [1,4,7,8]. Finally, this study focused on geographical effects by sampling around Guadeloupe Island in the north and around Martinique Island in the south. Among 60 species, this effect was significant for 25 species (42%). The condition factor of fish from Martinique Island was often higher than those from Guadeloupe Island. Consequently, at the north and south of Dominica islands, the environmental conditions could be different, with better conditions and growth of the fish in the south around Martinique Island than that noted for those around Guadeloupe Island. This geographical difference could be directly linked to habitat characteristics explained by environmental factors, such as temperature and salinity, but also to feeding activities (e.g., food availability and feeding rate) [4,6,7,8,26,45,46].

## 5. Conclusions

The length–weight relationships reported here for 109 commercial fish species from 24,996 individuals caught around Guadeloupe and Martinique Islands by sampling from October 2021 to September 2022, with potential spatial, temporal and sex differences, have yielded updated information for some species and provided new information on these relationships between biological parameters (total weight, total and fork lengths) for 16 species for the Atlantic Ocean, especially the Caribbean Sea. This study showed a moderate difference in size linked to sex, but a higher discrepancy appeared due to geographical and temporal effects. The sampling quarter (linked to the reproduction period), the sampling island, and sexual dimorphism showed significant effects on the LWR for 35, 25, and 24 species, respectively. This dataset was assembled with special French authorizations to analyze fish under the commercial size and individuals of species contaminated by ciguatera poisoning. These data and results will be essential for stock assessment of Caribbean fishes.

## Figures and Tables

**Figure 1 animals-13-03852-f001:**
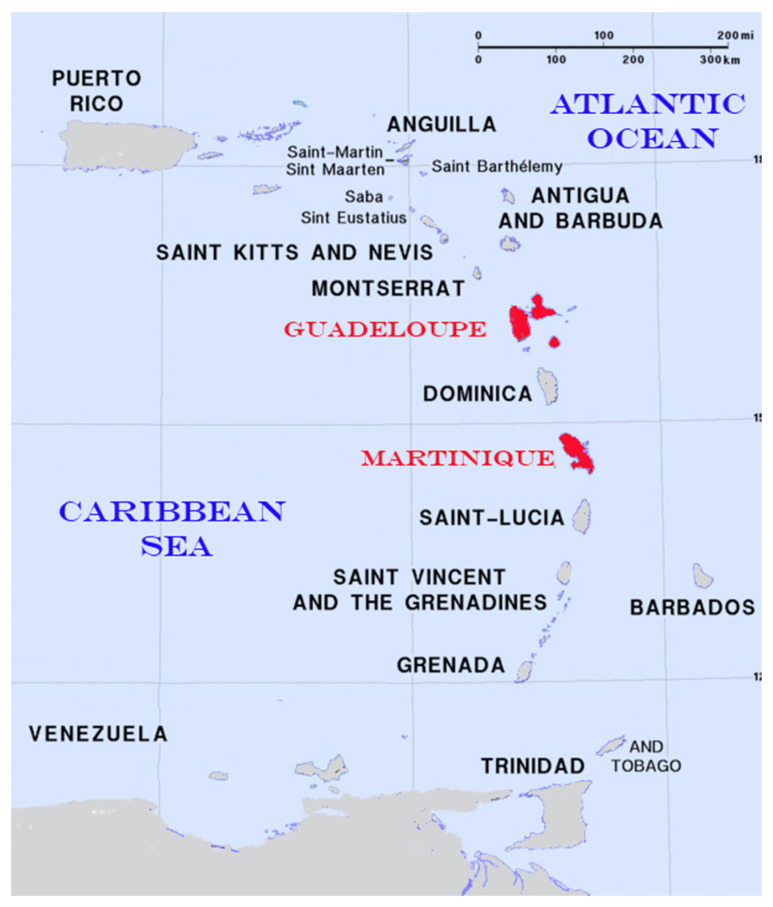
Map of the Lesser Antilles with the French West Indies (Guadeloupe and Martinique Islands) where all fish species were caught from September 2021 to August 2022.

**Figure 2 animals-13-03852-f002:**
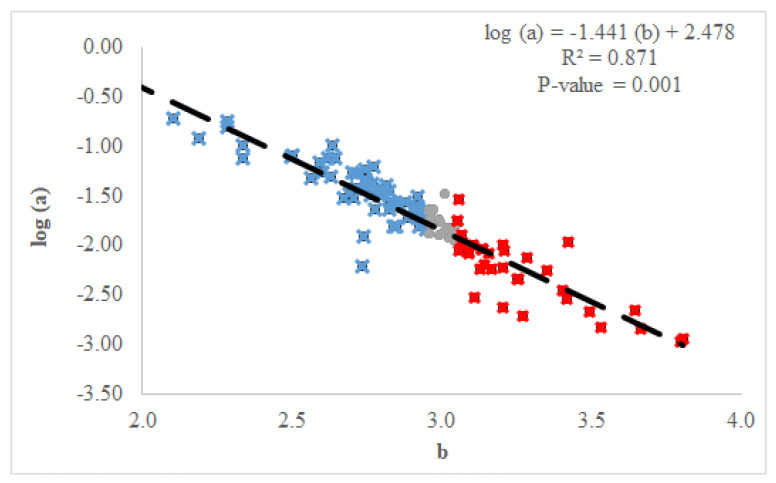
Plot of length–weight relationships (log a vs. b) in 109 fish species sampled from September 2021 to August 2022 around the French West Indies (Guadeloupe and Martinique Islands) (negative allometric growth with b < 3: blue cross; isometric growth with b = 3.0: grey circle; positive allometric growth with b > 3: red cross).

**Figure 3 animals-13-03852-f003:**
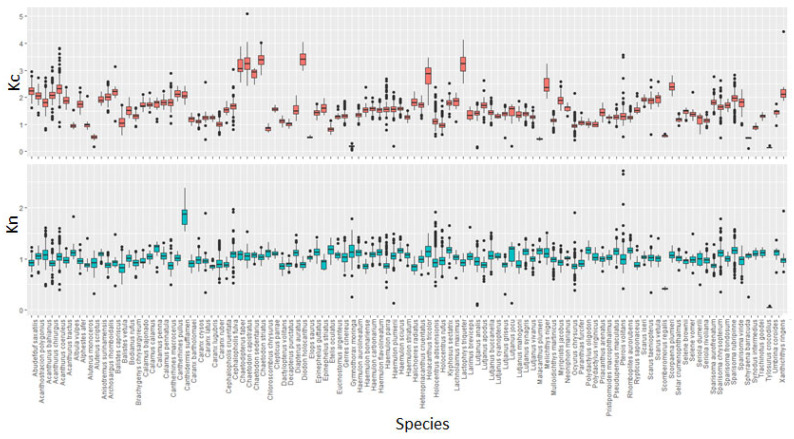
Boxplot of condition factors (Fulton’s factor: Kc; Relative condition factor: Kn) for each species sampled from September 2021 to August 2022 around the French West Indies.

**Figure 4 animals-13-03852-f004:**
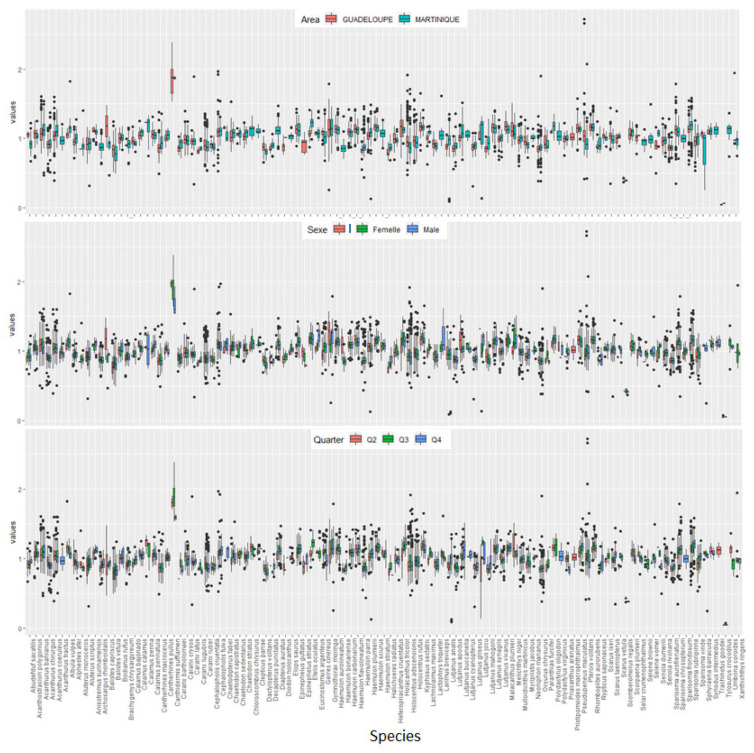
Boxplot of the relative condition factor (Kn) for each species sampled around the French West Indies according to the geographical location/sampling island, the sex (I for immature, female and male) and the sampling period (Q; Quarter 4 2021; Quarter 2 2022 and Quarter 3 2022).

**Table 1 animals-13-03852-t001:** Relationship between weight and body length for species with a specimen number greater than 30. Each species is identified (First Data with the star indicated that this information was presented for the first time), and this study is the first time that the relationship between two measurements was calculated with an individual number greater than 6. Here, *p*-values are reported for each LWR and LLR and for potential explanatory variables (sex, sampling quarter and area). The coefficients a and b of the LWR and the coefficients c and d of the LLR are presented, separately (value ± standard deviation). If the sample number was <5, no data are reported. Blue cells indicate significant *p*-values.

Family	Latin Name	N	W/TL							TL/FL			
			First Data	*p*-Value	a	b	Explanatory Factors			First Data	*p*-Value	c	d
							Sex	Quarter	Area				
Acanthuridae	*Acanthurus bahianus*	1187	*	<0.001	0.0222	2.9270	0.0645	<0.001	<0.001	*	<0.001	1.1532	−0.4007
	*Acanthurus chirurgus*	314		<0.001	0.0228	2.9702	0.2458	<0.001	0.0510		<0.001	1.0269	0.8089
	*Acanthurus coeruleus*	986		<0.001	0.0397	2.8139	0.1470	0.0498	<0.001		<0.001	1.1739	−0.9288
	*Acanthurus tractus*	99		<0.001	0.0341	2.7909	0.1870	no data	no data		<0.001	1.1564	−0.3920
Albulidae	*Albula vulpes*	74		<0.001	0.0057	3.1297	0.9329	0.0522	0.2420		<0.001	1.1570	0.6678
Balistidae	*Balistes vetula*	107		<0.001	0.1758	2.2849	0.1510	0.7050	0.0870		<0.001	1.4830	−3.3020
	*Melichthys niger*	62		<0.001	0.0570	2.7445	0.0163	<0.001	0.0130		<0.001	1.1289	−1.5303
Carangidae	*Caranx bartholomaei*	127		<0.001	0.0131	2.9573	0.9860	<0.001	<0.001		<0.001	1.2026	−0.3080
	*Caranx crysos*	273		<0.001	0.0303	2.7060	0.0182	<0.001	0.2920		<0.001	1.2414	−1.3354
	*Caranx latus*	203		<0.001	0.0188	2.8886	0.9980	0.6420	0.6440		<0.001	1.1715	0.6950
	*Caranx ruber*	907		<0.001	0.0056	3.1653	0.2457	<0.001	0.4613		<0.001	1.1614	0.7201
	*Selar crumenophthalmus*	47		<0.001	0.0035	3.4038	0.0050	<0.001	no data		<0.001	1.1146	0.1312
	*Seriola rivoliana*	404		<0.001	0.0082	3.0890	<0.001	<0.001	0.4230		<0.001	1.1138	1.0095
	*Chaetodon capistratus*	96		<0.001	0.0747	2.6228	0.7630	0.0570	0.9050		<0.001	1.1406	−0.9840
	*Chaetodon striatus*	87		<0.001	0.0328	3.0113	0.0300	0.1557	0.4154		<0.001	1.1131	−0.8654
Dactylopteridae	*Dactylopterus volitans*	82		<0.001	0.0045	3.2507	0.0239	0.0048	<0.001		<0.001	1.0623	−0.2675
Gerreidae	*Diapterus auratus*	370		<0.001	0.0494	2.6299	0.1454	0.0860	no data		<0.001	1.3193	−1.7101
	*Gerres cinereus*	439		<0.001	0.0233	2.8247	0.0040	0.2156	0.6586		<0.001	1.3045	−1.0524
Haemulidae	*Anisotremus surinamensis*	34		<0.001	0.0350	2.8270	0.7670	0.7550	no data		<0.001	1.1485	−0.4158
	*Brachygenys chrysargyreum*	261		<0.001	0.0015	3.5320	<0.001	<0.001	0.9056		<0.001	1.1581	−0.4070
	*Haemulon aurolineatum*	965		<0.001	0.0120	3.0448	0.0499	0.1806	0.3396		<0.001	1.0785	0.6476
	*Haemulon bonariense*	75		<0.001	0.0179	2.9584	0.0523	0.2718	no data		<0.001	1.0453	1.3000
	*Haemulon carbonarium*	133		<0.001	0.0269	2.8301	0.9411	0.9103	0.6760		<0.001	1.0924	0.5993
	*Haemulon flavolineatum*	444		<0.001	0.0181	2.9435	0.3211	<0.001	0.1764		<0.001	1.0424	1.2720
	*Haemulon parra*	290		<0.001	0.0251	2.8600	0.8203	<0.001	<0.001		<0.001	1.1174	−0.1802
	*Haemulon plumierii*	781		<0.001	0.0148	3.0160	0.3113	<0.001	<0.001		<0.001	1.1386	−0.3127
	*Haemulon sciurus*	131		<0.001	0.0263	2.8472	0.3704	0.6789	0.8320		<0.001	1.1144	−0.0610
	*Haemulon striatum*	230	*	<0.001	0.0152	2.9246	0.4230	0.7560	no data	*	<0.001	1.0880	0.7281
Holocentridae	*Holocentrus adscensionis*	1044		<0.001	0.0227	2.7794	<0.001	<0.001	<0.001		<0.001	1.0937	2.7445
	*Holocentrus rufus*	1189		<0.001	0.1205	2.1885	<0.001	0.1290	0.0160		<0.001	1.1037	2.3760
	*Myripristis jacobus*	93		<0.001	0.0269	2.8789	0.2940	0.0236	<0.001		<0.001	1.0512	1.5363
Kyphosidae	*Kyphosus sectatrix*	81	*	<0.001	0.0153	3.0448	0.2423	0.9156	0.6055		<0.001	1.1355	−0.2492
Labridae	*Bodianus rufus*	35	*	<0.001	0.0795	2.5023	no data	no data	0.9328		<0.001	1.2720	−4.7375
Lutjanidae	*Etelis oculatus*	87		<0.001	0.0153	2.8384	0.2012	0.4230	0.0940		<0.001	1.2478	−0.1504
	*Lutjanus analis*	223		<0.001	0.0106	3.0677	0.7902	0.6895	0.6997		<0.001	1.1310	−0.9373
	*Lutjanus apodus*	201		<0.001	0.0193	2.9615	0.2956	0.4244	0.0205		<0.001	1.0878	−0.5060
	*Lutjanus buccanella*	934		<0.001	0.0100	3.1056	0.0090	<0.001	0.0420		<0.001	1.0976	−0.3253
	*Lutjanus griseus*	223		<0.001	0.0158	2.9645	0.7357	0.8046	0.0459		<0.001	1.0737	−0.7727
	*Lutjanus mahogoni*	152		<0.001	0.0081	3.1582	0.7490	<0.001	<0.001		<0.001	1.0382	0.6515
	*Lutjanus synagris*	278		<0.001	0.0165	2.9424	0.1243	0.1829	0.1247		<0.001	1.0958	−0.5297
	*Lutjanus vivanus*	623		<0.001	0.0128	2.9912	<0.001	<0.001	0.0047		<0.001	1.1091	−0.4535
	*Ocyurus chrysurus*	1501		<0.001	0.0153	2.8503	0.1864	0.0217	<0.001		<0.001	1.2636	−1.0014
	*Rhomboplites aurorubens*	653		<0.001	0.0145	2.9466	0.7792	<0.001	0.0019		<0.001	1.1084	−0.1796
	*Aluterus scriptus*	49		<0.001	0.0019	3.2708	0.6717	0.2308	0.8046		no data	no data	no data
	*Cantherhines macrocerus*	272		<0.001	0.0056	3.3534	0.0014	0.0980	<0.001		no data	no data	no data
	*Cantherhines pullus*	34	*	<0.001	0.0362	2.8053	no data	no data	no data		no data	no data	no data
Mullidae	*Mulloidichthys martinicus*	543		<0.001	0.0085	3.0918	0.1891	0.0138	0.0002		<0.001	1.1587	0.4023
	*Pseudupeneus maculatus*	406		<0.001	0.0111	3.0419	0.0018	<0.001	0.3271		<0.001	1.0989	2.7970
Muraenidae	*Gymnothorax moringa*	44		<0.001	0.0061	2.7333	0.1225	0.6521	0.2816	*	<0.001	1.1000	0.0056
Ostraciidae	*Acanthostracion polygonius*	84		<0.001	0.0522	2.7156	0.0320	0.2361	0.7632	*	<0.001	1.0013	0.0143
	*Lactophrys triqueter*	72		<0.001	0.0622	2.7725	0.5302	0.4754	no data		<0.001	1.0045	0.0113
Pomacanthidae	*Holacanthus tricolor*	72		<0.001	0.0756	2.6444	0.0341	<0.001	0.6526		<0.001	1.1075	−1.2626
	*Abudefduf saxatilis*	93		<0.001	0.0417	2.7690	0.1196	0.7918	no data		<0.001	1.0355	1.6459
Priacanthidae	*Heteropriacanthus cruentatus*	70		<0.001	0.0322	2.7937	0.0581	0.6366	0.4549		<0.001	1.0264	−0.5200
	*Priacanthus arenatus*	85		<0.001	0.0772	2.4984	0.1452	0.0012	0.0091		<0.001	1.1112	−1.9655
Scaridae	*Scarus iseri*	39		<0.001	0.0248	2.9126	0.4427	no data	no data		<0.001	1.0218	−0.3942
	*Scarus taeniopterus*	329	*	<0.001	0.0101	3.2045	<0.001	0.0061	0.0022	*	<0.001	1.0198	−0.3683
	*Sparisoma aurofrenatum*	758	*	<0.001	0.0374	2.7505	0.0020	<0.001	0.6702	*	<0.001	1.0941	−1.3135
	*Sparisoma chrysopterum*	850		<0.001	0.0147	3.0299	<0.001	<0.001	0.9707		<0.001	1.1556	−1.9630
	*Sparisoma frondosum*	44	*	<0.001	0.0171	2.9970	no data	no data	no data	*	<0.001	1.1461	−2.0712
	*Sparisoma rubripinne*	1486		<0.001	0.0457	2.7454	<0.001	<0.001	<0.001		<0.001	1.0608	−1.0393
	*Sparisoma viride*	357		<0.001	0.0694	2.5944	<0.001	<0.001	<0.001		<0.001	1.2526	−4.3414
	*Umbrina coroides*	33		<0.001	0.0092	3.1329	no data	no data	no data		no data	no data	no data
Scorpaenidae	*Pterois volitans*	449		<0.001	0.0131	3.0009	0.8033	0.0001	<0.001		<0.001	1.0099	0.3873
	*Scorpaena plumieri*	49		<0.001	0.0306	2.9222	0.7582	0.0434	0.9689		<0.001	1.0078	0.5323
Serranidae	*Alphestes afer*	101		<0.001	0.0074	3.2861	0.2136	0.0055	0.2495		<0.001	1.0016	−0.0286
	*Cephalopholis cruentata*	60		<0.001	0.0322	2.7612	0.5269	0.9870	0.2416		<0.001	1.0100	0.1642
	*Cephalopholis fulva*	679		<0.001	0.0208	2.9266	0.9365	<0.001	0.1535		<0.001	1.0099	0.1584
	*Epinephelus guttatus*	220		<0.001	0.0092	3.1367	0.0235	<0.001	0.5750		<0.001	1.0989	0.3421
Sparidae	*Archosargus rhomboidalis*	362		<0.001	0.0230	2.9560	0.6310	0.1649	no data		<0.001	1.1433	−0.8086
	*Calamus calamus*	44		<0.001	0.0088	3.2120	0.8551	0.6276	no data		<0.001	1.1338	0.3011
	*Calamus pennatula*	131		<0.001	0.0236	2.9191	0.1427	<0.001	0.3989		<0.001	1.1229	0.3017

## Data Availability

The data presented in this study are available in Appendix A.

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
