# Peer review of "Morphometric Relationships between Length and Weight of 109 Fish Species in the Caribbean Sea (French West Indies)"

_animals, 2023, doi:10.3390/ani13243852_

Round 1

Reviewer 1 Report

Comments and Suggestions for Authors

The manuscript describes a large-scale study of the relationship between length and weight in a considerable number of primarily tropical fish species; for some species, such materials were obtained for the first time. An analysis of the possible effect on this relationship of a number of biological parameters such as the sexual composition of the population, living conditions, seasonality and a number of others was carried out. Based on the volume of material and partly on its analysis, the presented work deserves publication in the Animals journal. At the same time, there are a number of comments, primarily related to the presentation of the results obtained.

Judging by the material, the authors dealt with fish of commercial size, which is not reflected in the title or in the description of the materials. There is no doubt that the absence of younger ages affected the length and weight parameters for almost all fish species.

The manuscript is also clearly overloaded with tabular material. The main part contains three tables in which all 109 fish names are repeated. In addition to this, there are three more similar tables in the Supplements.  I would recommend to leave one table with the most important data that is primarily of scientific interest and characterizes the results obtained. 

Also I would recommend presenting a significant portion of the results worth discussing into graphs, similar to Figure 1. Or to refer the interested reader to the Supplements.

It would be appropriate to translate into graphic form, dividing into three groups (b = 3, b < 3, b > 3), the relationship between length and weight and discuss the results obtained for each of the groups separately. 

When mentioning species names for the first time, it is necessary to present the authors and years of description, as required by the Code of Zoological Nomenclature. 

The Conclusion seems rather vague, which is largely due to the lack of real discussion of the results obtained in the Discussion section. In particular, there is no mention in the Conclusion of the interesting result depicted in Figure 1.

With that said, the manuscript as a whole should be substantially shortened, primarily by removing or transferring repetitive or redundant information to the Supplements.

Author Response

Dear Reviewer 1, you find the answer for your request below :

The manuscript describes a large-scale study of the relationship between length and weight in a considerable number of primarily tropical fish species; for some species, such materials were obtained for the first time. An analysis of the possible effect on this relationship of a number of biological parameters such as the sexual composition of the population, living conditions, seasonality and a number of others was carried out. Based on the volume of material and partly on its analysis, the presented work deserves publication in the Animals journal. At the same time, there are a number of comments, primarily related to the presentation of the results obtained.

Judging by the material, the authors dealt with fish of commercial size, which is not reflected in the title or in the description of the materials. There is no doubt that the absence of younger ages affected the length and weight parameters for almost all fish species.

We clarified the data and we added more information: “All specimens were from landings of fishing boats with commercial and non-commercial species, covering all lengths, including below the legal size with the authorisation from the French government to realize the first large sampling for stock assessment. In the non-commercial fishes, there were discards species with low price and others contaminated by ciguatera poisoning. All specimens were weighed (one individual weight: total, WT ± 0.1 g) and measured (two individual lengths: total, TL ± 0.5 cm and fork, FL ± 0.5 cm). “

The manuscript is also clearly overloaded with tabular material. The main part contains three tables in which all 109 fish names are repeated. In addition to this, there are three more similar tables in the Supplements.  I would recommend to leave one table with the most important data that is primarily of scientific interest and characterizes the results obtained. 

We added the new analyses through the figures and finally there is only 1 table in version 2

Also I would recommend presenting a significant portion of the results worth discussing into graphs, similar to Figure 1. Or to refer the interested reader to the Supplements.

We presented two new figures from the Kc and Kn analysis and one figure of sampling area, and moved table 1 to supplementary. We also presented the map of lesser Antilles as recommended by the reviewer

It would be appropriate to translate into graphic form, dividing into three groups (b = 3, b < 3, b > 3), the relationship between length and weight and discuss the results obtained for each of the groups separately. 

It was done in the new figure 2

When mentioning species names for the first time, it is necessary to present the authors and years of description, as required by the Code of Zoological Nomenclature. 

We are agreed but there are 109 species and if you must added this nomenclature information, it will be in supplementary tables because all . We verified in many papers on the large list of species, there are only latin name to identify the species, we applied the same approach.  

The Conclusion seems rather vague, which is largely due to the lack of real discussion of the results obtained in the Discussion section. In particular, there is no mention in the Conclusion of the interesting result depicted in Figure 1.

We completed the discussion with the new figures to illustrate the difference.

With that said, the manuscript as a whole should be substantially shortened, primarily by removing or transferring repetitive or redundant information to the Supplements.

We modified the presentation through the figures, leaving only 1 table in the paper as we deleted 2 tables. For the English, my colleague Kirsteen MacKenzie, native English speaker, modified this second version.  

Best regards

kélig mahé

Reviewer 2 Report

Comments and Suggestions for Authors

The study devotes the length-weight relationships of fish species from waters of the French West Indies (the Caribbean Sea). It is useful and very informative study. The data set is rather impressive since it encompasses measurements of 4 996 individuals of 109 fish species. For several species, the relationship between length and weight has been reported for the first time. The results are clear and well discussed. The manuscript is overall well written and worth publishing.

Only the next point remained unclear to me. I understand that the data were log-transformed since equation (2) is given? If equation (2) is used to determine the parameters, then SD is determined for the (log a). The SD of the coefficient a can be determined by potentiation, but then it must be asymmetrical. That is, positive and negative deviations do not coincide. But in Table 1 only one value for SD is presented.

Perhaps LWR power functions were fitted to untransformed data using any algorithm for nonlinear least squares regression from the R statistical package? In this case, I suggest to delete the equation (2) for clarity? So, please clarify the situation and describe statistical methods in more detail.

Author Response

Dear Reviewer 2, you find the answer for your request below :

The study devotes the length-weight relationships of fish species from waters of the French West Indies (the Caribbean Sea). It is useful and very informative study. The data set is rather impressive since it encompasses measurements of 4 996 individuals of 109 fish species. For several species, the relationship between length and weight has been reported for the first time. The results are clear and well discussed. The manuscript is overall well written and worth publishing.

Only the next point remained unclear to me. I understand that the data were log-transformed since equation (2) is given? If equation (2) is used to determine the parameters, then SD is determined for the (log a). The SD of the coefficient a can be determined by potentiation, but then it must be asymmetrical. That is, positive and negative deviations do not coincide. But in Table 1 only one value for SD is presented.

To clarify, we deleted the SD value for the all coefficients a, b, c & d in all tables

Perhaps LWR power functions were fitted to untransformed data using any algorithm for nonlinear least squares regression from the R statistical package? In this case, I suggest to delete the equation (2) for clarity? So, please clarify the situation and describe statistical methods in more detail.

There are 2 approaches with power function from the data or linear function from log transformation of the raw data. These 2 statistical approaches were validated. Finally, we deleted the equation 2 and we clarified the statistical approach.

Best regards

kélig mahé

Reviewer 3 Report

Comments and Suggestions for Authors

The authors provided a valuable effort in characterizing the several species in the study area, providing such useful information for stock assessment like the LWR.

Otherwise, the presentation and analyses of data appeared lacking, as well as the discussions, which not highlights enough the novelty of a study also focused on some species never evaluated before and not explain enough some hypotheses. Actually, the manuscript seems more similar to a technical note than a research paper.

In my opinion, the limit of 10 individuals to include a species within this study is not enough. For such kind of analysis, a bigger number of individuals need to obtain robust results. At least, the minimum number for these analysesmay be20 individuals, particularly when a species showed a wide size range.

Moreover, in such kind of a study, the curves of LWR should be showed. I know 109 curves are very numerous to include in a paper, but they may be moved in Supplementary Materials. Particularly, it would be significant showing the graphs for species which are the first time to be analysed.

The Table 1, displaying the data, need accurate revision. Strange size range are showed. For instance, Neoniphon marianus displayed a weight rang 54-3033g versus a length range 15-20 cm. Looks like strange a difference about 3 kg respect to 5 cm in size.

The condition factor estimated by Fulton equation is generally considered as an approximate value, and estimations by Le Cren (1951), evaluated as more robust, are more common in literature (e.g. Brosset et al., 2017; Basilone et al., 2020; Ragheb, 2023). Le Cren needs of some data and assumption before calculations, however authors have not used this equation, despite such information are available (i.e., coefficients a and b from LWR, and allometric condiotion). Could the authors explain the preference for the Fulton index, please? Such information would be provided in the manuscript.

Discussions need an enrichment. The authors not highlighted enough the effort in publishing for the first time a study in these area involving all these species (as many as 109 species!), included 16 never evaluated before for the LWR. Moreover, some hypothesis is not well discussed using properly suggestion from literature, but only proposed very briefly. Similarly, the Conclusions looks more similar to an abstract than a summarizing of observed evidences, on the contrary this information are clearly included in the abstract.

Please, find some minor comments in the pdf attached. 

  Reference

Le Cren, E.D., 1951. The length-weight relationship and seasonal cycle in gonad weight and condition in the perch (Perca fluviatilis). J. Anim. Ecol. 20, 201–219.

Basilone, G., et al.,2020. Spawning ecology of the European anchovy (Engraulis encrasicolus) in the Strait of Sicily: linking variations of zooplankton prey, fish density, growth, and reproduction in an upwelling system. Prog. Oceanogr. 184:102330.

Brosset, P., et al., 2017. Spatio-temporal patterns and environmental controls of small pelagic fish body condition from contrasted Mediterranean areas. Progr. Oceanogr. 151, 149162.

Ragheb, 2023. Length-weight relationship and well-being factors of 33 fish species caught by gillnets from the Egyptian Mediterranean waters off Alexandria. The Egyptian Journal of Aquatic Research, 49 (3), 361-367.

Comments on the Quality of English Language

The English must be revised by a native speaker.

Author Response

Dear reviewer 3, you find the answer to your request :

The authors provided a valuable effort in characterizing the several species in the study area, providing such useful information for stock assessment like the LWR.

Otherwise, the presentation and analyses of data appeared lacking, as well as the discussions, which not highlights enough the novelty of a study also focused on some species never evaluated before and not explain enough some hypotheses. Actually, the manuscript seems more similar to a technical note than a research paper.

We modified the presentation of the results with the figures, we added Kn results, and we added some information in the discussion. Finally, we rewrote the conclusion section. 

In my opinion, the limit of 10 individuals to include a species within this study is not enough. For such kind of analysis, a bigger number of individuals need to obtain robust results. At least, the minimum number for these analyses may be20 individuals, particularly when a species showed a wide size range.

We are agreed with this text:

“The results are presented in two groups, with one for the species showing the higher precision in the analysis (i.e. specimen number > 30) and the other where only preliminary results are presented in supplementary tables due to the small number of specimens (i.e. number < 30).”

Moreover, in such kind of a study, the curves of LWR should be showed. I know 109 curves are very numerous to include in a paper, but they may be moved in Supplementary Materials. Particularly, it would be significant showing the graphs for species which are the first time to be analysed.

For the new species, we added the figure in the supplementary file and for other species, 109 is the complete species number but the raw data are also available in the supplementary information.

We added text in the discussion section and we added the new figure S1:

“Among these species, 8 species had a higher precision sampling number (n>30; Supplementary Figure S1). “

The Table 1, displaying the data, need accurate revision. Strange size range are showed. For instance, Neoniphon marianus displayed a weight rang 54-3033g versus a length range 15-20 cm. Looks like strange a difference about 3 kg respect to 5 cm in size.

We verified all species and corrected the mistake for Neoniphon marianus with weight range 54-104g versus a length range 15-20 cm

The condition factor estimated by Fulton equation is generally considered as an approximate value, and estimations by Le Cren (1951), evaluated as more robust, are more common in literature (e.g. Brosset et al., 2017; Basilone et al., 2020; Ragheb, 2023). Le Cren needs of some data and assumption before calculations, however authors have not used this equation, despite such information are available (i.e., coefficients a and b from LWR, and allometric condiotion). Could the authors explain the preference for the Fulton index, please? Such information would be provided in the manuscript.

We presented the “The Fulton’s condition factor (Kc) and the relative condition factor (Kn) [4] were applied to compare the observed total weight at the given length to the predicted total weight from the LWR (Equations 4 and 5):

Kc=100.WT/TL3

(4)

Kn=WT/(a.TLb)”

(5)

And we presented the new two figures from the Kc and Kn analysis, moved table 1 to the supplementary, and presented the map of lesser Antilles as recommended by the reviewer

We added this information in the discussion section

“In this study, a value of 3.0 (i.e. isometric growth) for the allometric coefficient b was observed for only 9 species (8.2% of the total number of species, Figure 2). The range value of b was between 2.0 and 4.0, confirming the previous results for the large number of species [38]. For this reason and the correlation between two coefficients of LWR, two condition factors were used to understand the sources of difference among LWR. Traditionally, the Fulton’s condition factor (Kc) based on isometric growth is applied, but the relative condition factor (Kn), defined by allometric growth  (b<3 or b>3), is more representative for the wild fish species. “

Discussions need an enrichment. The authors not highlighted enough the effort in publishing for the first time a study in these area involving all these species (as many as 109 species!), included 16 never evaluated before for the LWR. Moreover, some hypothesis is not well discussed using properly suggestion from literature, but only proposed very briefly. Similarly, the Conclusions looks more similar to an abstract than a summarizing of observed evidences, on the contrary this information are clearly included in the abstract.

Please, find some minor comments in the pdf attached. 

  Reference

Le Cren, E.D., 1951. The length-weight relationship and seasonal cycle in gonad weight and condition in the perch (Perca fluviatilis). J. Anim. Ecol. 20, 201–219.

Basilone, G., et al.,2020. Spawning ecology of the European anchovy (Engraulis encrasicolus) in the Strait of Sicily: linking variations of zooplankton prey, fish density, growth, and reproduction in an upwelling system. Prog. Oceanogr. 184:102330.

Brosset, P., et al., 2017. Spatio-temporal patterns and environmental controls of small pelagic fish body condition from contrasted Mediterranean areas. Progr. Oceanogr. 151, 149–162.

Ragheb, 2023. Length-weight relationship and well-being factors of 33 fish species caught by gillnets from the Egyptian Mediterranean waters off Alexandria. The Egyptian Journal of Aquatic Research, 49 (3), 361-367.

 Another points

L24 : this is just speculation, for now neither supported from literature in similar area or something like that... In my opinion, is not matter for the abstract

We deleted this part of the sentence

L56 The acronym has been already introduced

We deleted the acronym in line 56

L62 : Move here the proper reference

Done from line 64

L68 : I'd like to suggest to introduce a fig. showing the study area

We added the figure 1 with the map of lesser Antilles

L100 : There are some difference in the analysis of fishes from scientific survey and commercial fishery? Were the fishes measured fresh or after freezing? Please, add information about

We added more information : “All specimens were from landings of fishing boats with commercial and non-commercial species, covering all lengths, including below the legal size with the authorisation from the French government to realize the first large sampling for stock assessment. In the non-commercial fishes, there were discards species with low price and others contaminated by ciguatera poisoning. All specimens were weighed (one individual weight: total, WT ± 0.1 g) and measured (two individual lengths: total, TL ± 0.5 cm and fork, FL ± 0.5 cm). “

L109 : Why sex didn't determined for all the species?

We added more information : “Sex was determined by macroscopic gonad observation for 83 species, but this biological data was not observed for 26 species. All specimens were analysed fresh and, for 26 species, the gonad observation was too difficult to obtain the results with a high accuracy.”

Table 1 : Please, check the size ranges. In some species, appeared too big discrepancy between length and weight range. e.g., 3kg vs 5 cm

This table was sent to supplementary material; we modified the weight mistakes for Neoniphon marianus, Seriola dumerili, Synodus intermedius and Anisotremus surinamensis  

Line 138 Table S3 has been introduced before than Table S2

We modified this reference with the new table S1 and we verified the order

L154 It is not clear the meaning of these information, because of they are referred to different species. Probably, would be more illustrative reporting results about the smaller (range and/or mean) and bigger species

In many papers, the smallest and largest individuals are presented identifying the name of species each time. We kept the same presentation for the length and weight ranges.   

L174 English need to be revised by a native speaker. For instance, too many "showed" and "presented" in very few lines

Kirsteen MacKenzie, native English speaker and a colleague in my lab, has modified this second version after the modifications suggested by reviewers.

L261 : Please, argue it. I agree with such a hypothesis, however Authors must describe similar cases from literature already which induced them to this suggestion

We completed this sentence with :

“This geographical difference could be directly linked to habitat characteristics explained by environmental factors such as temperature and salinity, but also to feeding activities (e.g. food availability and feeding rate) [4, 6-8, 26, 40-41].”

L267 : It looks like an abstract more than conclusions. For instance, "This study shows the potential spatial, temporal and sex differences linked to the tested species.” In my opinion, Conclusions should include something as “this study showed moderate difference in size linked to sex, but higher discrepancy appeared due to geographical effect”

We modified the conclusion as :

“The length–weight relationships reported here for 109 commercial fish species from 24 996 individuals caught around Guadeloupe and Martinique Islands by sampling from October 2021 to September 2022, with potential spatial, temporal and sex differences, has updated this information for some species and provided new information on these relationships between biological parameters (total weight, total and fork lengths) for 16 species for the Atlantic Ocean, especially the Caribbean Sea. This study showed moderate difference in size linked to sex, but higher discrepancy appeared due to geographical and temporal effects. The sampling quarter (linked to the reproduction period), the sampling island, and the sexual dimorphism showed significant effects on the LWR respectively for 35, 25, and 24 species. This dataset was assembled with French special authorizations to analyze fish under the commercial size and individuals of species contaminated by ciguatera poisoning. These data and results will be essential for stock assessment of Caribbean fishes. 

Round 2

Reviewer 1 Report

Comments and Suggestions for Authors

The authors have mostly made the necessary changes, which makes it more readable and the information more concise. The addition of illustrative material also improved the manuscript. At the same time, I believe that the large number of taxa mentioned does not relieve authors from the need to cite in the scientific literature the authors and year of description. I leave it up to the editors of the journal to decide this issue. With the exception of this point, the authors have made notable changes and responded to my comments quite effectively, which allows me to recommend the manuscript for publication in the journal.